# Burden, risk factors and outcomes associated with adequately treated hypothyroidism in a population-based cohort of pregnant women from North India

**Neeta Dhabhai[1], Ranadip Chowdhury[1]\*, Anju Virmani[2], Ritu Chaudhary[1‡], Sunita Taneja[1], Pratima Mittal[3‡], Rupali Dewan[3‡], Arjun Dang[4‡], Jasmine Kaur[1‡], Nita Bhandari[1]**

1 Centre for Health Research and Development, Society for Applied Studies, New Delhi, India, 2 Department of Pediatrics, Max Smart Superspeciality Hospital, Saket, Rainbow Hospital, New Delhi, India, 3 Department of Obstetrics and Gynaecology, Vardhaman Mahavir Medical College and Safdarjung Hospital, New Delhi, India, 4 Dr Dangs Lab, New Delhi, India

⊙ These authors contributed equally to this work.
‡ RC, PM, RD, AD and JK also contributed equally to this work.
\* ranadip.chowdhary@sas.org.in

**Data Availability Statement:** All relevant data are within the paper and its Supporting Information files.

## Abstract

Hypothyroidism is the commonest endocrine disorder of pregnancy, with known adverse feto-maternal outcomes. There is limited data on population-based prevalence, risk factors and outcomes associated with treatment of hypothyroidism in early pregnancy. We conducted analysis on data from an urban and peri-urban low to mid socioeconomic population-based cohort of pregnant women in North Delhi, India to ascertain the burden, risk factors and impact of treatment, on adverse pregnancy outcomes- low birth weight, prematurity, small for gestational age and stillbirth. This is an observational study embedded within the intervention group of the Women and Infants Integrated Interventions for Growth Study, an individually randomized factorial design trial. Thyroid stimulating hormone was tested in 2317 women in early (9–13 weeks) pregnancy, and thyroxin replacement started hypothyroid (TSH ≥2.5mIU/mL). Univariable and multivariable generalized linear model with binomial family and log link were performed to ascertain risk factors associated with hypothyroidism and association between hypothyroidism and adverse pregnancy outcomes. Of 2317 women, 29.2% (95% CI: 27.4 to 31.1) had hypothyroidism and were started on thyroxin replacement with close monitoring. Overweight or obesity was associated with increased risk (adjusted RR 1.29, 95% CI 1.10 to 1.51), while higher hemoglobin concentration was associated with decreased risk (adjusted RR 0.93, 95% CI 0.88 to 0.98 for each g/dL) for hypothyroidism. Hypothyroid women received appropriate treatment with no increase in adverse pregnancy outcomes. Almost a third of women from low to mid socio-economic population had hypothyroidism in early pregnancy, more so if anemic and overweight or obese. With early screening and adequate replacement, adverse pregnancy outcomes may be avoided. These findings highlight the need in early pregnancy for universal TSH

**Funding:** The main trial was funded by the Biotechnology Industry Research Assistance Council (BIRAC), Department of Biotechnology, Government of India under the Grand Challenges India- All Children Thriving Initiative (GCIACT Ref No: BIRAC/GCI/0085/03/14-ACT) and the Bill & Melinda Gates Foundation, USA (Grant ID #OPP1191052). The funders has no role in study design, data collection and analysis, decision to publish or preparation of the manuscript.

**Competing interests:** The authors declare that they have no competing interests.

**Abbreviations:** CI, confidence interval; ARR, adjusted relative risk; BMI, body mass index; LBW, low birth weight; SGA, small for gestational age; TSH, thyroid stimulating hormone; WINGS, women and Infants integrated interventions for growth study; ATA, american thyroid association; SST, serum separator tubes; CMIA, chemiluminescent microparticle immunoassay; CV, coefficient of variation; RR: risk ratios.

screening and adequate treatment of hypothyroidism; as well as for attempts to reduce pre and peri-conception overweight, obesity and anemia.

**Clinical trial registration:** Clinical trial registration of Women and Infants Integrated Interventions for Growth Study Clinical Trial Registry–India, #CTRI/2017/06/008908; Registered on: 23/06/2017, (http://ctri.nic.in/Clinicaltrials/pmaindet2.php?trialid=19339&EncHid=&userName=society%20for%20applied%20studies).

# Background

The thyroid gland plays a key role in pregnancy homeostasis and metabolic adaptations important for fetal development, as well as supply of energy to the mother. It undergoes adaptive changes to meet the increased demand during pregnancy, and women with low reserves during preconception, frequently enter pregnancy in hypothyroid state. Hypothyroidism is the commonest endocrine disorder of pregnancy, and if not adequately treated, can result in adverse pregnancy outcomes- growth restriction, prematurity, low birth weight (LBW) and stillbirth [1, 2]. However, the symptoms are nonspecific and insidious; hence, diagnosis is usually missed or made much later due to delayed reporting of pregnancy [3].

Globally, hypothyroidism affects 3–5% of all pregnant women [4], however, the prevalence is higher in South Asian countries [5, 6]. In a Chinese study of 2899 pregnant women, the prevalence of hypothyroidism (TSH >3.93 mIU/L) in the first trimester was 10.9% [7]. Yadav et al, in a meta-analysis of observational studies, reported a pooled prevalence of 11.01% in pregnant women in India [8], however, only two of the 54 studies were community based, and the cut off levels of TSH used were not uniform, varying from 2.3–4.5 mIU/mL in all three trimesters of pregnancy. Data from secondary and tertiary hospital across nine states in India showed 13.1% prevalence in the first trimester, with a TSH cutoff of 4.5 mIU/mL [9].

The thyroid stimulating hormone (TSH) assay is a simple, sensitive, commonly used screening tool for thyroid dysfunction in pregnancy but is limited by lack of uniformly accepted reference ranges which vary according to laboratory reference levels, type of assay, and population heterogeneity. The new third generation assays have high functional sensitivity as recommended by the American Thyroid Association (ATA) and are a good tool for diagnosing primary thyroid dysfunction [10]. The National Guidelines in India (2014) recommend treatment aimed at maintenance of TSH <2.5 mIU/mL in the first trimester and <3 mIU/mL in the second and third trimester of pregnancy, similar to the guidelines of the Endocrine Society [11, 12]. The ATA 2017 recommends 4 mIU/mL as the upper limit of normal, in the absence of population specific ranges [13]. A systematic review of normative values of trimester specific thyroid function in Indian women concluded that TSH cut offs of up to 5–6 mIU/mL, similar to the pre-pregnancy stage, should be used in the first trimester of pregnancy, although it was limited by the fact that no outcomes were included [14].

There are limited data on population-based prevalence, risk factors and adverse outcomes of hypothyroidism in pregnancy, particularly in low to mid socio-economic populations. We conducted an analysis of data from a population-based cohort of urban and peri-urban low-mid socioeconomic strata neighborhoods of Delhi to determine the prevalence of hypothyroidism in pregnancy, its risk factors, and impact of replacement on adverse pregnancy outcomes LBW, prematurity, small for gestational age (SGA), and stillbirth.

## Material and methods

### Study design, setting and participants

This is an observational study embedded within the intervention group of the Women and Infants Integrated Interventions for Growth Study (WINGS), conducted in urban and peri-urban low-to-mid-socioeconomic neighborhoods in south Delhi, India. A summary of WINGS is provided below; the details of methods have been previously published [15]. Briefly, 13,500, eligible women aged 18–30 years were identified through a door-to-door survey. Women who provided written consent to participate in the study were enrolled (first randomization) to receive pre and peri-conception interventions or routine care and followed up until their pregnancies were confirmed, or for 18 months after enrollment. At confirmation of pregnancy by ultrasonography, after obtaining written consent a second randomization was done wherein pregnant women received enhanced pregnancy and early childhood care or routine care.

### Study description

Pregnant women in the intervention group received at least eight antenatal care visits. Body mass index (BMI) was measured, and a non-fasting serum TSH and hemoglobin tested at the time of confirmation of pregnancy. Women with TSH $\geq 2.5$ mIU/mL were labelled as hypothyroid and were managed with replacement doses of levothyroxine, after a thorough history and clinical assessment as per the National Guidelines [11]. TSH levels were repeated monthly, and thyroxine dose titrated till TSH stabilized to normal pregnancy reference levels [11]. Once normalized, a repeat TSH for monitoring, was done twice in the second trimester, and once in the late third trimester which amounted to an approximate five to six repeat assays of TSH in pregnancy. Close follow up was done by study team workers who made weekly home visits to ensure compliance, and progress was monitored using electronic trackers. Those with uncontrolled TSH and TSH $<0.1$ mIU/mL (hyperthyroid) were referred to the endocrinology clinic of the collaborating government tertiary care hospital (Safdarjung Hospital, New Delhi, India).

### Laboratory analysis

Blood samples for TSH assay were collected in serum separator tubes (SST), transported in cool boxes (4˚to 8˚C) to the field laboratory where centrifugation was done at ~450 × g at room temperature for 10 minutes, to separate serum and then transported to a National Accreditation Board for Testing and Calibration Laboratories (NABL) accredited study laboratory (Oncquest Laboratory) maintaining cold chain.

Serum TSH was analyzed using the Architect (Ci 8200 Abbott-Architect) TSH assay, a two-step immunoassay using chemiluminescent microparticle immunoassay (CMIA) technology with an inter-assay coefficient of variation (CV) of 20% which meets the requirements of a of third generation TSH assay. The Architect TSH assay has an analytical sensitivity of $<0.01$ μIU/mL [16].

### Ethics

The Ethics Review Committees of the Society for Applied Studies, Vardhman Mahavir Medical College and Safdarjung Hospital, and the World Health Organization, Geneva approved the study conducted with the relevant guidelines and regulations (e.g. Declaration of Helsinki). Written informed consent was obtained from the study participants.

## Statistical analysis

Sociodemographic characteristics were reported as mean (SD), or proportions, as appropriate. We calculated incidence (95% confidence interval: CI) of hypothyroidism occurring at the time of pregnancy confirmation. Univariable and multivariable generalized linear model with binomial family and log link were performed to ascertain risk factors associated with hypothyroidism. We calculated unadjusted and adjusted risk ratios (RR) and their 95% CI for the association between hypothyroidism and adverse pregnancy outcomes (LBW, SGA, preterm birth, spontaneous preterm birth, stillbirth) using generalized linear model with binomial family and log link. The candidate variables were related to socio-demographic and nutritional status of the women; continuous (maternal age, hemoglobin and glycosylated hemoglobin (HbA1c) levels at the time of confirmation of pregnancy), and categorical (height ($<150$ and $\geq150$ cm), years of schooling $<12$ and $\geq12$ years), early pregnancy (gestational age $\leq20$ weeks), BMI, religion (Hindu and others), type of family (extended or joint, and nuclear), family with a below-poverty-line card, and family covered by health insurance scheme. All statistical analyses were performed using STATA version 16 (StataCorp, College Station, TX, USA).

## Definitions of adverse pregnancy outcomes

LBW was be defined as weight $< 2500$ g on day 7 after birth. Gestation at birth was estimated by subtracting date of birth from date of dating ultrasound and adding it to gestational age as assessed by dating ultrasound according to INTERGROWTH-21 [17]. Preterm birth was defined as births occurring at $< 37$ completed weeks of gestation. Spontaneous preterm births will be defined as births occurring at $< 37$ weeks of gestation and preterm pre-labor rupture of membranes or spontaneous onset of labor. Still birth was defined as babies born with no signs of life at or after 28 weeks of gestation, 1000 grams or more, or attainment of at least 35 cm crown-heel length (WHO Maternal, newborn, child, and adolescent health).

Birth weight centile was calculated using the INTERGROWTH-21 standard based on day-7 weight and gestational age at birth. SGA was defined as birth weight centile $< 10^{th}$ as per INTERGROWTH-21 standard.

## Results

In this study 2317 women were followed up from pregnancy till delivery. The median (IQR) of gestational age at the time of recruitment was 9.5 (9.1–11.0) weeks. Socioeconomic and clinical characteristics of enrolled women prior to pregnancy are shown in Table 1. Women who were hyperthyroid (TSH<0.1mIU/ml) were excluded in this analysis. The study population was relatively young with a mean (SD) age of 23.8 (3.1) years, about half of them had education more than 12 years. Height was less than 150 cm in 34.1%; mean (SD) BMI was 22.2 (3.9) kg/m$^2$; with dual burden of malnutrition: 15% women were underweight and 28% of women were overweight or obese in the hypothyroid group compared to 22% in the euthyroid group (Table 1).

Table 2 shows the thyroid status of pregnant women at the time of confirmation of pregnancy, with 29.2% (95% CI: 27.4 to 31.1) having hypothyroidism (TSH $\geq2.5$ mIU/mL).

Table 3 shows the association between socioeconomic and clinical characteristics of women with hypothyroidism. Overweight or obesity in early pregnancy was associated with increased risk (adjusted RR 1.29, 95% CI 1.10 to 1.51) of hypothyroidism. Each unit increase in Hb (adjusted RR 0.93, 95% CI 0.88 to 0.98 for each g/dL) levels was associated with reduced risk of being hypothyroidism.

Table 4 shows the association of hypothyroidism with adverse pregnancy outcomes where management of hypothyroidism was supported by the research team. The risk of adverse

**Table 1. Sociodemographic characteristics of pregnant women.**

| Characteristics of pregnant women | Euthyroid n = 1576 | Hypothyroid n = 677 |
|---|---|---|
| Age (years), mean (SD) | 23.7 (3.1) | 24.0 (3.0) |
| Height (cm), mean (SD) | 152.4 (5.5) | 152.2 (5.8) |
| Height <150 cm | 527 (33.4) | 246 (36.3) |
| Schooling ≥12 yr | 805 (51.1) | 332 (49.0) |
| Homemaker | 1503 (95.4) | 644 (95.1) |
| Early pregnancy BMI, mean (SD) | 22.3 (3.9) | 22.9 (4.1) |
| Underweight (<18.5 kg/m$^2$) | 242 (15.4) | 91 (13.4) |
| Overweight or obesity (≥25 kg/m$^2$) | 352 (22.3) | 192 (28.4) |
| Hindu | 1288 (81.7) | 566 (83.6) |
| Family had a below poverty line card | 52 (3.3) | 23 (3.4) |
| Family covered by a health insurance scheme | 161 (10.2) | 78 (11.5) |
| Joint or extended family* | 1032 (65.5) | 458 (67.6) |

All values are numbers (percentages) unless stated otherwise

* Joint or extended family: Adult relatives other than the enrolled woman's husband and children living together in a household

pregnancy outcomes i.e., LBW, SGA, preterm, spontaneous preterm birth and stillbirth, were similar among euthyroid women and those who were treated for hypothyroidism.

## Discussion

The main findings of this study were, the high prevalence of hypothyroidism in early pregnancy, occurring in 29.2% of the population-based cohort and that treating hypothyroidism, early and adequately led to no increase in adverse outcomes (stillbirth, preterm birth, LBW, SGA). Anemia, overweight and obesity in early pregnancy were identified as risk factors for hypothyroidism.

A wide disparity in the prevalence of hypothyroidism in pregnancy has been reported in previous studies in India. A hospital-based study from nine states of India reported 15% prevalence of hypothyroidism using TSH levels of >4.5 mIU/mL and of 44.3% using a cut off level of >2.5 mIU/mL in first trimester [9]. A meta-analysis of observational studies with varying gestational ages and TSH cut off values found the prevalence of hypothyroidism in pregnancy to be 11% [8]. A prospective observational study from central part of India found hypothyroidism in 9.1% in the third trimester of pregnancy using a cut off of TSH >5 mIU/mL. Similar to our study, Bein et al in a systemic review and metaanlysis reported a decrease in risk of pregnancy loss and neonatal death with treatment of subclinical hypothyroidism [18].

Similar to our finding, a study from southern part of India showed overweight or obesity was a risk factor for hypothyroidism in early pregnancy, and high maternal TSH is associated

**Table 2. Thyroid status at the time of confirmation of pregnancy.**

| TSH level | n(%) | 95% CI | Mean (SD) | Median (IQR) |
|---|---|---|---|---|
| | n = 2317 | | | |
| TSH <0.1 mIU/mL | 64 (2.8) | 2.1 to 3.5 | 0.04 (0.02) | 0.04 (0.04 to 0.05) |
| TSH 0.1 to <2.5 mIU/mL | 1576 (68.0) | 66.1 to 69.9 | 1.3 (0.6) | 1.3 (0.8 to 1.8) |
| TSH ≥2.5 to <10 mIU/mL | 652 (28.1) | 26.3 to 30.0 | 4.0 (1.4) | 3.6 (3.0 to 4.6) |
| TSH ≥ 10 mIU/mL | 25 (1.1) | 0.6 to 1.6 | 51.9 (124.2) | 15.4 (13.0 to 33.6) |

**Table 3. Potential risk factors for hypothyroidism in enrolled women.**

| Risk factors for hypothyroidism | Unadjusted RR (95% CI) | Adjusted RR (95% CI) |
|---|---|---|
| Age (per 1 year) | 1.02 (0.99 to 1.04); p = 0.057 | 1.01 (0.99 to 1.04); p = 0.227 |
| Height (<150 cm) | 1.09 (0.96 to 1.24); p = 0.182 | 1.05 (0.91 to 1.21); p = 0.517 |
| Schooling ≥12 yr | 0.94 (0.83 to 1.07); p = 0.375 | 0.92 (0.80 to 1.06); p = 0.241 |
| Homemaker | 0.96 (0.72 to 1.29); p = 0.802 | 1.06 (0.75 to 1.50); p = 0.741 |
| Joint or extended family | 1.07 (0.93 to 1.23); p = 0.321 | 1.11 (0.95 to 1.29); p = 0.176 |
| Family had below-poverty-line card | 1.02 (0.72 to 1.44); p = 0.905 | 1.02 (0.70 to 1.48); p = 0.921 |
| Family covered by health insurance scheme | 1.10 (0.90 to 1.33); p = 0.348 | 1.07 (0.86 to 1.33); p = 0.541 |
| Hindu religion | 1.10 (0.92 to 1.31); p = 0.291 | 1.10 (0.91 to 1.33); p = 0.310 |
| Early pregnancy BMI | | |
| Normal BMI | Reference | Reference |
| Underweight (<18.5 kg/m$^2$) | 0.95 (0.79 to 1.16); p = 0.637 | 0.99 (0.80 to 1.22); p = 0.912 |
| Overweight or obesity (≥25 kg/m$^2$) | 1.23 (1.07 to 1.42); p = 0.004 | 1.29 (1.11 to 1.52); p = 0.001 |
| Hemoglobin at pregnancy confirmation (per 1gm/dL) | 0.96 (0.91 to 1.01): p = 0.088 | 0.93 (0.88 to 0.98): p = 0.007 |
| HbA1c (%) at pregnancy confirmation (per 1 percentage) | 0.96 (0.80 to 1.15); p = 0.629 | 0.91 (0.77 to 1.10); p = 0.328 |

with obesity and higher weight gain [19, 20]. The association of obesity and overweight with adverse outcomes is well documented including a recent systemic review and metaanalysis [21, 22]. Similarly, our finding of a higher hemoglobin level associated with reduced risk of hypothyroidism, was echoed in a systemic review and meta analysis which found that iron deficiency adversely affects thyroid function and autoimmunity in pregnant women [19, 23]. Also, a study from China found that proportion of hypothyroidism was higher in women with mild anemia in the first trimester than in women with no anemia [24].

**Table 4. Association of management of hypothyroidism with adverse pregnancy outcomes compared to subjects who were euthyroid.**

| Outcome | Euthyroid | Hypothyroid | Unadjusted RR | Adjusted RR* |
|---|---|---|---|---|
| | n(%) | n(%) | (95% CI) | (95% CI) |
| Low birth weight | n = 1329 | n = 560 | 1.09 | 1.09 |
| | 309 (23.3) | 142 (25.4) | (0.92 to 1.30) | (0.92 to 1.30) |
| Small for gestational age | n = 1376 | n = 586 | 1.19 | 1.20 |
| | 353 (25.7) | 180 (30.7) | (1.03 to 1.39) | (0.98 to 1.44) |
| Preterm birth | n = 1592 | n = 649 | 0.92 | 0.90 |
| | 195 (12.8) | 76 (11.7) | (0.72 to 1.18) | (0.70 to 1.15) |
| Spontaneous preterm birth | n = 1529 | n = 649 | 0.77 | 0.78 |
| | 117 (7.7) | 38 (5.9) | (0.54 to 1.10) | (0.54 to 1.11) |
| Stillbirth | n = 1540 | n = 659 | 1.56 | 1.54 |
| | 21 (1.4) | 14 (2.1) | (0.80 to 3.04) | (0.79 to 3.01) |

*adjusted for maternal age, height, years of schooling, early pregnancy (gestational age ≤20 weeks) BMI

The strengths of our study are that it is population-based; the socioeconomic strata is low to middle income, which represents the average Indian population; serum TSH was used for thyroid status assessment, which is feasible as a screening test in settings like India where pregnant women may not return after the first visit due to economic reasons, lack of easy accessibility to the health center, etc., and the evidence that early identification and adequate treatment, reduced the risk of adverse pregnancy outcomes. The limitations of our study are that we did not test for other thyroid hormones (T4, T3, Free T4, Free T3), so we could not distinguish subclinical from overt hypothyroidism; nor did we test for thyroid antibodies, which could have helped us detect "at risk" pregnancies.

This study has important implications for health care of women of the reproductive age group. First, the high burden of hypothyroidism in early pregnancy and the prevention of adverse pregnancy outcomes with early management highlights the need for early detection and treatment in antenatal care programs. Secondly, preventive interventions could be introduced preconceptionally to reduce risk factors, like obesity and overweight, and low hemoglobin levels.

## Conclusion

Almost a third of women from a mid-low socio-economic population may be at risk for developing hypothyroidism in early pregnancy. Anemia, being overweight or obese may increase the risk. Ensuring universal TSH screening in early pregnancy, and adequate treatment if hypothyroidism is detected, would serve to improve pregnancy outcomes. Reducing pre and periconception obesity and anemia could help reduce the risk of hypothyroidism.

## Supporting information

**S1 Dataset.**
(XLSX)

## Acknowledgments

We the Society for Applied Studies gratefully acknowledge the contribution and support of the mothers who participated in the study, their families and others in the community. We acknowledge the core support provided by the Department of Maternal, Newborn, Child and Adolescent Health, World Health Organization, Geneva (WHO Collaborating Centre IND-158).

## Author Contributions

**Conceptualization:** Neeta Dhabhai, Ranadip Chowdhury, Anju Virmani, Sunita Taneja, Nita Bhandari.

**Formal analysis:** Neeta Dhabhai, Ranadip Chowdhury, Anju Virmani, Sunita Taneja, Nita Bhandari.

**Investigation:** Neeta Dhabhai.

**Visualization:** Ranadip Chowdhury, Anju Virmani, Nita Bhandari.

**Writing – original draft:** Neeta Dhabhai, Ranadip Chowdhury, Anju Virmani, Sunita Taneja, Nita Bhandari.

**Writing – review & editing:** Neeta Dhabhai, Ranadip Chowdhury, Anju Virmani, Ritu Chaudhary, Sunita Taneja, Pratima Mittal, Rupali Dewan, Arjun Dang, Jasmine Kaur, Nita Bhandari.

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
