## [Decision Letter · Decision Letter 0]

6 Apr 2023

PONE-D-23-04138Burden, risk factors and outcomes associated with adequately treated hypothyroidism in a population-based cohort of pregnant women from North IndiaPLOS ONE

Dear Dr. Chowdhury,

Thank you for submitting your manuscript to PLOS ONE. After careful consideration, we feel that it has merit but does not fully meet PLOS ONE’s publication criteria as it currently stands. Therefore, we invite you to submit a revised version of the manuscript that addresses the points raised during the review process.

We look forward to receiving your revised manuscript.

Kind regards,

Jing Zhang

Academic Editor

PLOS ONE

Reviewers' comments:

Reviewer's Responses to Questions

**Comments to the Author**

1. Is the manuscript technically sound, and do the data support the conclusions?

Reviewer #1: Yes

Reviewer #2: Partly

2. Has the statistical analysis been performed appropriately and rigorously? 

Reviewer #1: Yes

Reviewer #2: Yes

3. Have the authors made all data underlying the findings in their manuscript fully available?

Reviewer #1: Yes

Reviewer #2: No

4. Is the manuscript presented in an intelligible fashion and written in standard English?

Reviewer #1: Yes

Reviewer #2: Yes

5. Review Comments to the Author

Reviewer #1: The manuscript addresses an important issue and presents findings from analysis of data collected from a population-based cohort of pregnant women in North Delhi, India. The authors have attempted to ascertain the burden, risk factors and impact of treatment, on adverse pregnancy outcomes. The methods and the findings have been presented well and I recommend publication. However, I have some suggestions that I would like the authors to consider.

1. I would suggest that the authors should focus on the associations observed and refrain from translating them to causality. With this nature of analysis, it may be difficult to ascertain causality. For instance, they have mentioned that higher haemoglobin levels were protective. This could be written as: “Each unit increase in Hb levels was associated with reduced risk of being hypothyroid”.

2. Lines 138-141: I would suggest it to write differently. For instance, the heading of Table 4 should be modified to : Table 4. Association of management of hypothyroidism with pregnancy outcomes, compared to subjects who were euthyroid. On similar lines, please modify the statements in 138-140: The risk of adverse pregnancy outcomes i.e., LBW, SGA, preterm, spontaneous preterm birth and stillbirth, were similar among those with no hypothyroidism and those who were treated for hypothyroidism”.

3. Please use the terms consistently- the authors have used “Euthyroid” and “no hypothyroid” interchangeably.

4. I could note from Table 2 that around 3% of the women studied were “hyperthyroid” i.e., TSH <0.1 mIU/mL. I am surprised to see that they have been grouped under “Euthyroid” (in Table 1). I would suggest that all the analysis be re-done after excluding women with TSH levels <0.1 mIU/mL.

Reviewer #2: This well written and sound paper from Ranadip Chowdhury and colleagues presents data from an observational study embedded within the Women and Infants Integrated Interventions for Growth Study, an individually randomized factorial design trial. The study was conducted to ascertain the burden, risk factors and impact of treatment, on adverse pregnancy outcomes- low birth weight, prematurity, small for gestational age and stillbirth. My comments primarily relate to clarifications, and suggestions for additional information regarding the statistical analyses.

1. Line 100 - First mention of NABH should be in full and NABH in parenthesis

2. The authors should consider giving a bit more detail on definitions of the adverse pregnancy outcomes such as detail on how SGA was determined e.g. SGA (<10th percentile weight for gestational age using INTERGROWTH 21 standards)

3. Table 3 - It would be good if the authors can give an indication of how much missing data there was and any assumptions on the missing data mechanisms

4. Table 3 - It is not very clear which variables where adjusted for in the adjusted analyses

5. Table 3 - Under Hemoglobin at pregnancy confirmation, "per 1 percentage" should read "for each d/dL" ?

6. PLOS authors have the option to publish the peer review history of their article (what does this mean?). If published, this will include your full peer review and any attached files.

Reviewer #1: No

Reviewer #2: No

---

## [Author Response · Author response to Decision Letter 0]

12 Apr 2023

Point-to-point rebuttal letter for the revised manuscript

“Burden, risk factors and outcomes associated with adequately treated hypothyroidism in a population-based cohort of pregnant women from North India”

Response: We have checked the PLOS ONE’s style requirements and ensured the same including file naming. 

Response: Thank you. We have provided the correct information.

The main trial was funded by the Biotechnology Industry Research Assistance Council (BIRAC), Department of Biotechnology, Government of India under the Grand Challenges India- All Children Thriving Initiative (GCIACT Ref No: BIRAC/GCI/0085/03/14-ACT) and the Bill & Melinda Gates Foundation, USA (Grant ID #OPP1191052).

The funding agencies did not play any role in study design and are neither involved in nor have any influence over the collection, analyses or interpretation of data.

Response: We have uploaded the minimal anonymized data set necessary to replicate the study findings as supporting information titled S1 Data Set.

Response: We have included the following ethics statement in the Methods section (Lines 109 to 112). 

The Ethics Review Committees of the Society for Applied Studies, Vardhman Mahavir Medical College and Safdarjung Hospital, and the World Health Organization, Geneva approved the study conducted with the relevant guidelines and regulations (e.g. Declaration of Helsinki). Written informed consent was obtained from the study participants.

5. Please review your reference list to ensure that it is complete and correct. If you have cited papers that have been retracted, please include the rationale for doing so in the manuscript text or remove these references and replace them with relevant current references. Any changes to the reference list should be mentioned in the rebuttal letter that accompanies your revised manuscript. If you need to cite a retracted article, indicate the article’s retracted status in the References list and also include a citation and full reference for the retraction notice.

Response: We have added the following references in the introduction section number 5, 6 and 17, 18 under definitions of adverse pregnancy outcomes. No reference has been retracted.

5. Rajput R, Goel V, Nanda S, Rajput M, Seth S. Prevalence of thyroid dysfunction among women during the first trimester of pregnancy at a tertiary care hospital in Haryana. Indian journal of endocrinology and metabolism. 2015;19(3):416.

6. Sletner L, Jenum AK, Qvigstad E, Hammerstad SS. Thyroid function during pregnancy in a multiethnic population in Norway. Journal of the Endocrine Society. 2021;5(7):bvab078.

17. TGHN. International Fetal and Newborn Growth Consortium for the 21st Century, or INTERGROWTH-21st.

18. WHO. Stillbirth. fact sheet.

Reviewer #1: The manuscript addresses an important issue and presents findings from analysis of data collected from a population-based cohort of pregnant women in North Delhi, India. The authors have attempted to ascertain the burden, risk factors and impact of treatment, on adverse pregnancy outcomes. The methods and the findings have been presented well and I recommend publication. However, I have some suggestions that I would like the authors to consider.

Response: Thank you. 

1. I would suggest that the authors should focus on the associations observed and refrain from translating them to causality. With this nature of analysis, it may be difficult to ascertain causality. For instance, they have mentioned that higher haemoglobin levels were protective. This could be written as: “Each unit increase in Hb levels was associated with reduced risk of being hypothyroid”.

Response: Thank you for the suggestion. We have revised as following relevant sections (Lines 155 and 156).

Each unit increase in Hb (adjusted RR 0.93, 95% CI 0.88 to 0.98 for each g/dL) levels was associated with reduced risk of being hypothyroidism 

2. Lines 138-141: I would suggest it to write differently. For instance, the heading of Table 4 should be modified to: Table 4. Association of management of hypothyroidism with pregnancy outcomes, compared to subjects who were euthyroid. On similar lines, please modify the statements in 138-140: The risk of adverse pregnancy outcomes i.e., LBW, SGA, preterm, spontaneous preterm birth and stillbirth, were similar among those with no hypothyroidism and those who were treated for hypothyroidism”.

Response: Thanks for the suggestions. We have revised as following the title of Table 4 (Lines 164 to 166) and subsequently the interpretation of table 4 (Lines 160 to 162) in the revised manuscript. 

Lines 164 to 166 

Table 4 Association of management of hypothyroidism with adverse pregnancy outcomes compared to subjects who were Euthyroid 

Lines 160 to 162

The risk of adverse pregnancy outcomes i.e., LBW, SGA, preterm, spontaneous preterm birth and stillbirth, were similar among euthyroid women and those who were treated for hypothyroidism

3. Please use the terms consistently- the authors have used “Euthyroid” and “no hypothyroid” interchangeably.

Response: Thank you for this important observation, we have now used the term “Euthyroid” consistently. 

4. I could note from Table 2 that around 3% of the women studied were “hyperthyroid” i.e., TSH <0.1 mIU/mL. I am surprised to see that they have been grouped under “Euthyroid” (in Table 1). I would suggest that all the analysis be re-done after excluding women with TSH levels <0.1 mIU/mL.

Response: Thanks for the suggestions. We have re-done all analyses (Table 1, 3, 4) excluding women with TSH levels <0.1 mIU/mL in the revised manuscript. The revised estimates are in the similar lines. 

Reviewer #2: This well written and sound paper from Ranadip Chowdhury and colleagues presents data from an observational study embedded within the Women and Infants Integrated Interventions for Growth Study, an individually randomized factorial design trial. The study was conducted to ascertain the burden, risk factors and impact of treatment, on adverse pregnancy outcomes- low birth weight, prematurity, small for gestational age and stillbirth. My comments primarily relate to clarifications, and suggestions for additional information regarding the statistical analyses.

1. Line 100 - First mention of NABH should be in full and NABH in parenthesis

Response: Thank you. We have mentioned as following the corrected full form (Line 102). 

National Accreditation Board for Testing and Calibration Laboratories (NABL)

2. The authors should consider giving a bit more detail on definitions of the adverse pregnancy outcomes such as detail on how SGA was determined e.g. SGA (<10th percentile weight for gestational age using INTERGROWTH 21 standards)

Response: Thanks for the suggestions. We have included as following the definitions of the adverse pregnancy outcomes in the methods section (Lines 127-135). 

LBW was be defined as weight < 2500 g on day 7 after birth, Gestation at birth was estimated by subtracting date of birth from date of dating ultrasound and adding it to gestational age as assessed by dating ultrasound according to INTERGROWTH-21. Preterm birth was defined as births occurring at < 37 completed weeks of gestation. Spontaneous preterm births will be defined as births occurring at < 37 weeks of gestation and preterm pre-labor rupture of membranes or spontaneous onset of labor. Still birth was defined as babies born with no signs of life at or after 28 weeks of gestation, 1000 grams or more, or attainment of at least 35 cm crown-heel length (WHO Maternal, newborn, child, and adolescent health. Birth weight centile was calculated using the INTERGROWTH-21 standard based on day-7 weight and gestational age at birth. SGA was defined as birth weight centile < 10th as per INTERGROWTH-21 standard. 

3. Table 3 - It would be good if the authors can give an indication of how much missing data there was and any assumptions on the missing data mechanisms

Response: We have only 1 missing data for all the variables included in the model for adjusted relative risks (Table 3) except HbA1c level for which ~10% data were missing. The estimates were in the similar if we exclude HbA1c from the final model. 

We did not have any assumptions on the missing data mechanism. We included the pregnant women whose all variables were present. 

4. Table 3 - It is not very clear which variables where adjusted for in the adjusted analyses

Response: The following variables were adjusted:

Continuous (maternal age, hemoglobin and glycosylated hemoglobin (HbA1c) levels at the time of confirmation of pregnancy), and categorical (height (<150 and ≥150 cm), years of schooling <12 and ≥12 years), early pregnancy (gestational age ≤20 weeks), BMI, religion (Hindu and others), type of family (extended or joint, and nuclear), family with a below-poverty-line card, and family covered by health insurance scheme.

5. Table 3 - Under Hemoglobin at pregnancy confirmation, "per 1 percentage" should read "for each d/dL"?

Response: Thank you for raising this point. Yes, we agree the correct unit for hemoglobin should be “for each gm/dL”. We have revised accordingly in the revised manuscript.

---

## [Decision Letter · Decision Letter 1]

16 Aug 2023

Burden, risk factors and outcomes associated with adequately treated hypothyroidism in a population-based cohort of pregnant women from North India

PONE-D-23-04138R1

Dear Dr. Ranadip,

We’re pleased to inform you that your manuscript has been judged scientifically suitable for publication and will be formally accepted for publication once it meets all outstanding technical requirements.

Kind regards,

Surangi Jayakody, MBBS, MSc, MD

Academic Editor

PLOS ONE

Additional Editor Comments (optional):

Reviewers' comments:

Reviewer's Responses to Questions

**Comments to the Author**

1. If the authors have adequately addressed your comments raised in a previous round of review and you feel that this manuscript is now acceptable for publication, you may indicate that here to bypass the “Comments to the Author” section, enter your conflict of interest statement in the “Confidential to Editor” section, and submit your "Accept" recommendation.

Reviewer #2: All comments have been addressed

Reviewer #3: All comments have been addressed

2. Is the manuscript technically sound, and do the data support the conclusions?

Reviewer #2: Yes

Reviewer #3: Yes

3. Has the statistical analysis been performed appropriately and rigorously? 

Reviewer #2: Yes

Reviewer #3: Yes

4. Have the authors made all data underlying the findings in their manuscript fully available?

Reviewer #2: Yes

Reviewer #3: Yes

5. Is the manuscript presented in an intelligible fashion and written in standard English?

Reviewer #2: Yes

Reviewer #3: Yes

6. Review Comments to the Author

Reviewer #2: (No Response)

Reviewer #3: I would like to express my sincere appreciation for your exceptional dedication and hard work throughout this project. Your efforts have been commendable. I would also like to extend my gratitude for promptly addressing the previous comments raised by the reviewers. Your revisions have significantly improved the clarity and quality of the paper.

To ensure the paper's comprehensibility for readers, I kindly request that you address the following queries:

1- The title: Could you kindly provide an explanation for choosing the term "Adequately Treated Hypothyroidism" instead of "Inadequately Treated Hypothyroidism"? This clarification would help readers better understand the rationale behind your choice of terminology.

2- Were there any other associated risk factors observed in the recruited patients during pregnancy? It would be beneficial to include information about any additional factors that may have influenced the outcomes for these patients.

3- It would be valuable to include the gestational ages at which the patients were diagnosed with hypothyroidism and whether there were any variations in outcomes based on the gestational age. For instance, did the timing of hypothyroidism detection, particularly in cases of late treatment, have any impact on the outcomes?

4- In Table 4, you compare the euthyroid group to the hypothyroid group who received treatment. However, no statistical significance is highlighted for adverse outcomes in the hypothyroid group without treatment. Could you please clarify the reasons for not indicating any statistical significance in this particular comparison?

5- Since the focus is on comparing adverse pregnancy outcomes between the euthyroid group and the hypothyroid group receiving treatment, it would be beneficial to address the limitations that prevented tracking the outcomes of the untreated hypothyroid group. Additionally, please clarify whether the hypothyroid group receiving treatment underwent controlled thyroid function tests.

6- Regarding the clinical trial registration, the provided registration (Clinical Trial Registry – India, #CTRI/2017/06/008908; Registered on: 23/06/2017) appears to be the same registration for a different study published by some of the same authors (PMID: 35031013). Could you please provide an explanation or clarification for this similarity in registration details?

7. PLOS authors have the option to publish the peer review history of their article (what does this mean?). If published, this will include your full peer review and any attached files.

Reviewer #2: No

Reviewer #3: **Yes: **Mena Abdalla

---

## [Editor Report · Acceptance letter]

4 Sep 2023

PONE-D-23-04138R1 

Burden, risk factors and outcomes associated with adequately treated hypothyroidism in a population-based cohort of pregnant women from North India 

Dear Dr. Chowdhury:

I'm pleased to inform you that your manuscript has been deemed suitable for publication in PLOS ONE. Congratulations! Your manuscript is now with our production department. 

Kind regards, 

on behalf of

Dr Surangi Jayakody 

Academic Editor

PLOS ONE